# Multi-Omics Interpretation of Anti-Aging Mechanisms for ω-3 Fatty Acids

**DOI:** 10.3390/genes12111691

**Published:** 2021-10-24

**Authors:** Shu-Hui Xie, Hui Li, Jing-Jing Jiang, Yuan Quan, Hong-Yu Zhang

**Affiliations:** 1College of Informatics, Huazhong Agricultural University, Wuhan 430070, China; xieshuhui@webmail.hzau.edu.cn (S.-H.X.); lihui2611@webmail.hzau.edu.cn (H.L.); jiangjj@webmail.hzau.edu.cn (J.-J.J.); 2Hubei Key Laboratory of Agricultural Bioinformatics, College of Informatics, Huazhong Agricultural University, Wuhan 430070, China; zhy630@mail.hzau.edu.cn

**Keywords:** ω-3 fatty acids, aging, methylation, Mendelian randomization, gut microbiome

## Abstract

Aging is one of the hottest topics in biomedicine. Previous research suggested that ω-3 fatty acids have preventive effects on aging. However, most of previous studies on the anti-aging effects of ω-3 fatty acids are focused on clinical observations, and the anti-aging mechanisms of ω-3 fatty acids have not been fully elucidated. This stimulated our interest to use multi-omics data related to ω-3 fatty acids in order to interpret the anti-aging mechanisms of ω-3 fatty acids. First, we found that ω-3 fatty acids can affect methylation levels and expression levels of genes associated with age-related diseases or pathways in humans. Then, a Mendelian randomization analysis was conducted to determine whether there is a causal relationship between the effect of ω-3 fatty acids on blood lipid levels and variation in the gut microbiome. Our results indicate that the impact of ω-3 fatty acids on aging is partially mediated by the gut microbiome (including *Actinobacteria, Bifidobacteria* and *Streptococcus*). In conclusion, this study provides deeper insights into the anti-aging mechanisms of ω-3 fatty acids and supports the dietary supplementation of ω-3 fatty acids in aging prevention.

## 1. Introduction

Aging is a natural physiological phenomenon for living organisms, which refers to gradual degenerative changes and increased frailty of the body with increasing age [1]. Due to the fact that biological functions of cells are performed by proteins, senescence may (in part) be the result of imbalance/dysfunction of the cellular proteome (protein homeostasis). Aging is characterized by the accumulation of cell damage, which results in increased susceptibility relative to complex diseases such as cancer, type 2 diabetes and cardiovascular diseases. The root of these diseases lies in the aging process itself, and aging is the highest risk factor for the development of these complex diseases. Effectively inhibiting aging is one of the important solutions to prevent aging-related diseases and achieve longevity [2]. Therefore, anti-aging is one of the hottest topics in biomedicine. However, whether it is for anti-aging or treating aging-related diseases, individuals usually need long-term medication, which can cause the drug resistance of humans. Compared with drug prevention, dietary supplementation is safer and more feasible.

Currently, a large number of studies have indicated that dietary supplementation of ω-3 fatty acids has an effect on preventing aging [3]. Ω-3 fatty acids, also known as N-3 fatty acids, are a class of unsaturated fatty acids. The important ω-3 essential fatty acids include α-linolenic acid (ALA), eicosapentaenoic acid (EPA) and docosahexaenoic acid (DHA), all of which are polyunsaturated fatty acids. Studies have found that ω-3 fatty acids may have the potential to prevent and reduce the complications of aging [4,5], including cognitive decline and cardiovascular disease. The levels of EPA, DHA and total ω-3 PUFAs in the peripheral blood tissues of dementia patients are significantly lower [6]. DHA, which is abundant in the brain, is neuroprotective and helps in maintaining proper brain function. The brain concentration of DHA is determined by its dietary DHA content and hepatic conversion from dietary derived ALA, as verified by an in vivo experimental model in which the brain incorporation rate of DHA is equal to the brain consumption rate of DHA [7]. In a cohort of 1214 older non-demented people monitored for four years, higher plasma EPA levels were linked to a lower incidence of dementia, regardless of depressive condition. Dietary deficiency of ω-3 fatty acids may result in accelerated brain aging, atrophy, partial memory and cognitive losses, resulting in high risks for Alzheimer’s disease and other dementia symptoms [8].

The primary sources of ω-3 fatty acids are Marine fatty fish (salmon, tuna, mackerel, herring, saury, halibut and sardines), krill, seeds (walnuts, flaxseeds, chia seeds, sesame seeds, pumpkin seeds and soybeans), certain leafy green vegetables, avocado and certain types of seaweed. The human body cannot synthesize ω-3 fatty acids de novo. Still, it can use the 18-carbon ω-3 fatty acids, i.e., α-linolenic acid (ALA), as the raw material, prolong the carbon chain through the human body’s enzymes to synthesize the 20-carbon unsaturated ω-3 fatty acids (EPA) and then synthesize the 22-carbon unsaturated ω-3 fatty acids (DHA) from EPA [8]. Due to the limited ability of humans to extend and desaturate α-linolenic acid into long-chain ω-3 polyunsaturated fatty acids, it is necessary to obtain adequate amounts through fish and fish oil products containing high levels of ω-3 polyunsaturated fatty acids [3].

As people age, the body’s ability to synthesize DHA from ALA decreases. As a result, DHA deficiency may be present in the elderly [3]. However, most of the studies on the effect of ω-3 fatty acids on aging at this stage are focused on clinical observations, and the underlying anti-aging mechanisms of ω-3 fatty acids have not been fully elucidated. In this study, based on multi-omics data, we performed methylation analysis, transcriptome analysis and Mendelian randomization analysis to interpret the anti-aging mechanisms of ω-3 fatty acids.

## 2. Materials and Methods

### 2.1. Methylation Analysis

In this study, methylation chip data are from the GEO database (https://www.ncbi.nlm.nih.gov/geo/query/acc.cgi?acc=GSE89278, accessed on 28 September 2021). The data are from a double-blind randomized placebo-controlled trial in which pregnant mothers consumed DHA-rich fish oil (800 mg DHA/d) or placebo supplements from 20 weeks gestation to delivery. Blood dots were collected from the children at birth (*n* = 991), and the researchers examined the overall DNA methylation of all the samples. Genome methylation data at birth from 369 children were obtained: 179 for the control group and 190 for the experimental group.

Then, two methods were used for differential methylation analysis: one is based on a Statistical difference of DNA Methylation between Promoter and Other Body Region (SIMPO) algorithm, and the other is based on the traditional method of methylation probe annotation.

Differential methylation analysis based on SIMPO algorithm: In methylation microarray detection, multiple methylation probes are distributed in the functional regions of the same gene, and different probes measure different values of methylation signals. Due to the randomness and complexity of genomic DNA methylation, it is too one-sided to consider only the methylation of the promoter region or only the methylation of the gene body region, and there is no significant correlation between the two. A study found that the correlation between the difference in DNA methylation between the functional region and the promoter region of a gene and the gene expression was as high as 0.67 [9], and based on this finding, Quan et al. developed the SIMPO algorithm that can extract the correlation between the DNA methylation level of a gene and its expression level [10]. It has been proved that this method resulted in significant improvements in gene overlaps (from 5 to 17%) between different datasets, and the robustness of SIMPO is better than the traditional probe-based method. In addition, the biological significance of phenotype-related genes identified by the SIMPO algorithm is comparable to that of the traditional probe-based methods [10]. The SIMPO algorithm can count the differences in DNA methylation between the promoter and other gene body regions and convert the methylation levels of multiple probes into the methylation levels of the genes, expressed by SIMPO scores.

The formula of the SIMPO algorithm is as follows:$$SIMPO\;score=\frac{\bar{x}-\bar{y}}{S_{w}\sqrt{\frac{1}{m}+\frac{1}{n}}}\sim t\left(m+n-2\right)$$
where the following is the case.
$$S_{w}=\frac{1}{m+n+1}\left[\left(m-1\right)S_{1}^{2}+\left(n-1\right)S_{2}^{2}\right]$$

$\bar{x\;}$ is the average methylation value of all probes in the gene body region; $\bar{y}$ is the average methylation value of all probes in the promoter region; m is the number of probes in the gene body region; n is the number of probes in the promoter region; $S_{1}^{2}$ is the variance of the methylation value of the probe in the gene body region; and $S_{2}^{2}$ is the variance of the methylation value of the probe in the promoter region. $S_{w}$: The variance of the difference of average methylation value and average gene body value.

After obtaining the gene SIMPO scores of the control group and the fish oil supplemented experimental group, a *t*-test was performed. The value of 0.05 was used as the threshold of *p*-value to screen the differentially methylated genes.

Differential methylation analysis based on methylation probe annotation: The R package “minfi” [11] was used to read and filter the probes from the original methylation data, and the “DMPFinder” function was used to analyze the differential methylation of the matrix. The “type” parameter was set to “categorical”. From the results of differential methylation analysis, significant sites were screened (Q-value < 0.001), and differential methylation genes were obtained by site annotation.

Next, we took the intersection of the differential genes obtained by the two methods and looked up their functional annotations on PubMed.

Then Gene Ontology(GO) enrichment analysis was performed using STRING (https://string-db.org/, accessed on 28 September 2021). The differential methyl-ation genes were then entered into the STRING website for GO analysis. Finally, Kyoto En-cyclopedia of Genes and Genomes (KEGG) enrichment analysis was conduct-ed using KOBAS (http://kobas.cbi.pku.edu.cn/, accessed on 28 September 2021). Appendix A show the detailed process results of Methylation analysis.

### 2.2. Transcriptome Analysis

In this study, the gene expression profile data were from the GEO database (https://www.ncbi.nlm.nih.gov/geo/query/acc.cgi?acc=GSE50945, accessed on 3 August 2021). Data were obtained from human placenta HTR8/SVneo cells. The samples contained 12 array culture experiments. Four were blank controls, four were treated with DNA-rich fish oil emulsion and four were treated with soybean oil emulsion. Cell culture conditions are as follows: 200,000 HTR8/SVneo cells per well were seeded in SRM in a 6-well plate and incubated for 24 h at 37 °C in 20% oxygen and 5% CO_2_. After 24 h, 2 mL of either SRM (N), SRM containing 50 μM Soy Oil emulsion (S) or SRM containing 50 μM Fish Oil emulsion (D) was added to each well. In this study, four samples treated with DNA-rich fish oil were used as the experimental group, and four samples treated with soybean oil emulsion and four samples untreated were combined as the control group.

The GEO2R analysis tool of the GEO website was used to perform differential expression analysis, and significant genes were screened (adj. *p*-value < 0.05). Then, the online analysis tool DAVID (https://david.ncifcrf.gov/home.jsp, accessed on 3 August 2021) was used for GO enrichment. After the analysis, gene functional annotation and functional enrichment results were produced. Then, the DAVID website was used for gene ID conversion. For each gene, OFFICAIL_GENE_SYMBOL was converted to ENTREI_GENE_ID, which was later submitted to the KOBAS for KEGG analysis and visualization to determine the metabolic pathways of these differential genes. Appendix A show the detailed process results of Transcription analysis.

### 2.3. A Two-Sample Mendelian Randomization Analysis

In Mendel randomization, the question of whether there was a causal relationship between the effect of fish oil on blood lipid level and the impact of the gut microbiome was discussed. The original data of the gut microbiome were NG16 and MBG. NG16 is the German population data published in 2016 with around 1800 people. MBG is the summary statistics for 152 microbial trait genome-wide association analyses. However, as an exposure variable, it is required to have a *p*-value of less than 5 × 10^−8^, among which NG16 does not meet the requirement; thus, the exposure factors used in MR analysis are only Single-nucleotide polymorphism(SNP) sites in MBG for which its *p*-value is less than 5 × 10^−8^ (six data pieces). MBG data were obtained from the University of Bristol Data Repository (https://doi.org/10.5523/bris.22bqn399f9i432q56gt3wfhzlc, accessed on 25 June 2021). It provides 211 gut microbiome and human SNPs loci GWAS information files. Lipid data with fish oil supplementation were obtained as outcome variables from the article [12]. There were more than 500,000 Caucasian volunteers recruited from 2006 to 2010 in England, Scotland and Wales; their biochemical, clinical and genotypic data were collected. All participants were between 40 and 70 years old when they were assessed.

Then, the R package “TwoSampleMR” [13] was used for Mendelian randomization analysis. In this part, Inverse Variance Weighting (IVW), Weighted Median (WM) and MR Egger regression were used as the main causal effect estimations. Among them, the IVW method focuses on integrating multiple genetic variants to infer the causal relationship between exposure factors and outcome variables. It is a weighted linear regression of Walder ratios calculated from different SNPs to obtain the causal relationship of unbiased estimation. WM is a weighted density function for ratio estimation. Multiple genetic effect values can be integrated by assigning different weight values to each genetic variable. If at least half of the information in the analysis comes from valid instrumental variables, then the causal effects can be continuously estimated. MR Egger takes gene pleiotropy into account, and the principle is the same as for IVW, but the difference is that the intercept term for IVW must be 0. In contrast, it is not necessarily the case for MR Egger, where the intercept is used to check whether there exist genetic pleiotropy. When the intercept corresponds to a statistic with a *p*-value > 0.05, it means that the causal effect is not influenced by genetic pleiotropy. The advantage of comparing the results from three different approaches is higher reliability due to increased consistency. A flowchart summarizing the two-steps diagram is presented in Figure 1.

Then, a gene pleiotropy test was conducted. One of the assumptions of MR analysis is that instrumental variables can only affect outcomes through exposure. If instrumental variables directly affect outcomes without affecting exposure, it violates the MR idea. Therefore, it is necessary to test whether there is genetic pleiotropy in causal inference between exposure and outcome. MR Egger regression analysis can be used to evaluate the bias caused by gene pleiotropy for the first step (effect of fish oil on lipid and gut microbiome variation) and the second step (effect of the gut microbiome variation on lipid) of the two-step MR mediation method, and its regression intercept can be used to evaluate the size of pleiotropy. The closer the intercept is to 0, the less the possibility of gene pleiotropy. In this study, the existence of gene pleiotropy in the analysis was measured by judging the *p*-value of the gene pleiotropy test. If *p* > 0.05, the possibility of gene pleiotropy in the causal analysis was considered to be weak, and its influence could be ignored.

In addition to using the above 3 methods (Inverse Variance Weighting, Weighted Median and MR Egger regression) to test the reliability and stability of the results, our study also adopted the leave-one-out method for sensitivity analysis. That is, the gut microbiome with a *p*-value less than 0.05 in the IVW method and that passed the heterogeneity test and gene pleiotropy test was removed one by one, and the combined effect of the remaining SNP was calculated to evaluate the influence of each SNP on the gut microbiome.

## 3. Results

### 3.1. Differential Methylation Analysis

In this study, differential methylation analysis of the methylation chips was performed by two methods. The first method is the following: The SIMPO algorithm and the SIMPO scores of control samples were subjected to *t*-test with the SIMPO scores of experimental group samples. Genes numbering 233 with significant differences in methylation levels were obtained (*p*-value < 0.05) for subsequent analysis. Then, GO and KEGG pathway analyses were performed on these genes, and the GO results are as follows: Molecular Function (MF) enrichment results showed four molecular functions (Table 1), all of which pointed to sequence-specific DNA binding functions, especially transcriptional regulation of specific regions of sequence-specific DNA binding; thus, it was hypothesized that supplementation of ω-3 fatty acids could affect transcriptional regulatory functions. After that, KEGG analysis of 233 differentially methylated genes was conducted using KOBAS. Two KEGG terms were obtained from the category of “KEGG_PATHWAY” with a *p*-value threshold of 0.005: 1. Signaling pathways regulating pluripotency of stem cells; 2. Cellular senescence. Three KEGG terms were obtained from the category of “KEGG_DISEASE” with a *p*-value threshold of 0.005: 1. Parkinsonian syndrome; 2. Lissencephaly; 3. Maturity onset diabetes of the young (MODY). Table 2 shows the detailed data of the KEGG analysis results. Figure 2 shows the bar chart and the bubble chart of the KEGG enrichment analysis results. The second method is outlined as follows: Seeking the differentially methylated probes between the experimental group and the control group and then using annotation information to find the differential Methylation genes corresponding to the probes. One hundred and sixty five significant differential methylation probes (DMPs) were obtained, and 140 differential methylated genes were acquired after annotation.

Then, we entered all 140 genes into the STRING website for GO enrichment analysis, and four biological GO terms that linked to aging have been discovered: 1. Homophilic cell adhesion via plasma membrane adhesion molecules; 2. Regulation of neurotransmitter levels; 3. Nervous system development; 4. Multicellular organism development. Table 3 shows the detailed data of the GO analysis results.

There is an intriguing GO enrichment result that 10 of 140 genes we entered were annotated in a biological process of regulation of neurotransmitter levels (Figure 3).

Then, the intersection of 140 and 233 differentially methylated genes obtained by the two methods used in the methylation analysis was obtained, and two genes were obtained as *UNC13A* and *OTOF* genes. Therefore, we could be more confident in speculating that *UNC13A* and *OTOF* genes show changes in methylation levels in response to supplementation of ω-3 fatty acids, both of which are upregulated. *UNC13A* and *OTOF* genes are both clinically reversible, but the mechanism of disease reversal is still unclear. Genome-level gene modification may be one of the explanations for the reversible nature of the disease.

Moreover, *UNC13A* and *OTOF* are included in the ten annotated genes. Both the *UNC13A* and *OTOF* genes are associated with neuropathy. *OTOF*: Mutations in this gene are a cause of neurosensory nonsyndromic recessive deafness. The homology suggests that this protein may be involved in vesicle membrane fusion. Several transcript variants encoding multiple isoforms have been found for this gene (https://www.ncbi.nlm.nih.gov/gene/9381, accessed on 28 September 2021). *UNC13A*: This gene encodes a member of the UNC13 family. *UNC13* proteins play important roles in neurotransmitter release at synapses [14]. Single nucleotide polymorphisms in this gene may be associated with sporadic Amyotrophic lateral sclerosis (ALS) (https://www.ncbi.nlm.nih.gov/gene/23025, accessed on 28 September 2021).

### 3.2. Transcriptome Analysis

In this study, a total of 891 differentially expressed genes (adjusted *p*-value < 0.05) were obtained by differential expression analysis between four experimental groups treated with DHA-rich fish oil and eight control groups that were not treated with DHA-rich fish oil. A volcano map was drawn (Figure 4), where the blue part shows down-regulated genes and the red part shows up-regulated genes. After removing some unannotated genes, 793 differentially expressed genes were acquired.

Then, we performed functional enrichment analysis of differentially expressed genes in DAVID. GAD_DISEASE, GO and KEGG databases were used to conduct the enrichment analysis of the differentially expressed genes. Significant outcomes related to aging were summarized in Table 4. In the GAD_DISEASE database, two diseases probably associated with aging were found: Alzheimer’s disease and longevity. According to GO analysis results, one Cellular Component probably associated with aging was found: mitochondria. As for KEGG, two pathways probably associated with aging were found: p53 and FoxO signaling pathways, which are closely related to cell senescence. The KOBAS website was used to visualize the KEGG enrichment analysis results. Figure 5 shows the bar chart and the bubble chart of the KEGG enrichment analysis results.

### 3.3. Mendelian Randomization Causal Association Analysis

In this section, the two-step Mendelian randomization (2-step MR) method was used to explore the causal relationship between the effect of fish oil on blood lipid levels and the influence of variation in the gut microbiome. SNP was used as the genetic instrumental variable. The gut microbiome we studied included *Actinobacteria*, *Bifidobacteria* and *Streptococcus*, and the lipid types we studied included high-density lipoprotein (HDL), low-density lipoprotein (LDL) and triacylglyceride (TAGs). The analysis consists of the following three evaluation models: Inverse-Variance Weighted (IVW), Weighted Median (WM) and MR Egger regression. The specific MR results are shown in Table 5, Table 6 and Table 7. The result of the *p*-value that is less than 0.05 indicates that the two have a causal relationship. The tables show the significant results. The corresponding scatter plots are shown as Figure 6, Figure 7 and Figure 8. For each gut microbiome, we chose one plot as an example. The OR value shows the risk of raising blood lipids. An OR < 1 means that for each standard deviation increase in a nutrient, the corresponding risk is reduced by a corresponding percentage of the OR value and vice versa.

According to the results in the table, we can infer that fish oil acts in part on three types of the gut microbiome, *Bifidobacteria, Streptococcus* and *Actinobacteria*, thus causing changes in the levels of LDL and TAGs. In addition, there is evidence that these three types of gut microbiome may have a strong association with the aging of hosts.

## 4. Discussion

In our study, we used three bioinformatics methods (i.e., differential methylation analysis, transcriptome analysis and Mendelian randomization analysis) to interpret the anti-aging mechanisms of ω-3 fatty acids.

Analysis of the influence of ω-3 fatty acids on DNA methylation: Biological functions and diseases related to early development include neurosensory nonsyndromic recessive deafness caused by *OTOF* mutations, Lissencephaly and Nervous system development. The biological processes and diseases associated with aging include Cellular senescence, Parkinsonian syndrome and ALS. Human aging is closely related to diseases and the development of the nervous system. In the analysis of the effect of supplementing fish oil rich in ω-3 fatty acids on gene expression, we found the molecular mechanism behind the development of the nervous system by ω-3 fatty acids. It is worth proposing that the supplementation of ω-3 fatty acids may have a therapeutic effect on ALS by modifying the methylation level of the *UNC13A* gene. This discovery has certain significance to further support the role of epigenetic modification in human developmental programming and to point the direction for future research. 

Protocadherin (Pcdh) cluster regulation has complex gene regulation mechanisms that are important for the normal development of the nervous system. Pcdh cluster genes encode calcium-related transmembrane proteins, which are mainly expressed in the nervous system. These neural adhesion proteins most likely play a critical role in the establishment and function of specific cell–cell connections in the brain (https://www.ncbi.nlm.nih.gov/gene/5097, accessed on 28 September 2021). A large number of tandem repeats are present in Pcdh clusters throughout vertebrate evolution. The transition of the repeated 5′ Pcdh-αc2 from conformably expressed to random expression in the wild-type brain, accompanied by increased DNA methylation, suggests that the tandem replication and methylation modifications in the Pcdh cluster can broaden and modify the function of gene types [15]. We speculated that the experimental conditions of ω-3 fatty acids supplementation in this study could modify the methylation of Pcdh cluster genes; thus, it has a positive effect on the development process of the nervous system. It is noteworthy that among the GO enrichment results, 10 of the 140 genes (or proteins) we imported were annotated in the biological GO Terms biological function network, among which *OTOF* and *UNC13A* were included in the 10 genes. Both *UNC3A* and *OTOF* genes are associated with neuropathy.

Amyotrophic lateral sclerosis (ALS) is a disease that results in the gradual deterioration and loss of function in the brain and spinal motor neurons, which eventually results in paralysis [16]. ALS of the prevalence and incidence increased with age [17]. The pathogenesis of ALS worldwide included an average age of 62. The prognosis for survival in patients with ALS is 2 to 5 years [16]. Symptoms of ALS may have a quite large reversal within a few weeks to months, but the “reversal” of ALS symptoms is often fleeting [18], and there is no reliable treatment for ALS. It is worth noting that a case-control study by Harrison et al. described 36 patients with ALS classified as “ALS reversals” from the Pooled Resource Open-Access ALS Clinical Trials (PRO-ACT) database; when compared with the control group, patients with ALS reversal were more likely to take curcumin, copper, azathioprine, fish oil, vitamin D and glutathione [18]. In previously existing studies, it was known that ω-3 fatty acids have positive effects on nervous system development [19,20]; thus, we hypothesized that ω-3 fatty acids in fish oil may have a positive effect in treating ALS through modifying the *UNC13A* gene’s methylation levels. This inference may guide treatment regimens for ALS disease.

Both classes of neurological diseases involved in the *UNC13A* and *OTOF* genes are reversible in clinical presentation [16,21], and the methylation levels of *UNC13A* and *OTOF* both had variability in the controlled trial of ω-3 fatty acids supplementations. Although the mechanism of the above disease reversal has not been defined, we reasoned that the methylation modification of the DNA gene by supplementing fish oil may be one of the causes or manifestations of neurological reversal.

In the transcriptome analysis, it can be observed from the differential gene enrichment results of samples treated with ω-3 fatty acids and unprocessed samples that these differential genes are significantly enriched in the genes associated with Alzheimer’s disease and longevity in the GAD_DISEASE database, enriched in mitochondria in GO and enriched in the p53 signaling pathway and FoxO signaling pathway in KEGG. These two pathways are linked to the aging process. The p53 transcription factor is important for cellular stress responses. When activated in response to DNA damage, it causes cell growth to stop, allowing DNA repair to take place, or it causes cellular senescence or apoptosis, preserving genome integrity [22]. The role of p53 in regulating cellular growth induced by strong oncogenic signals or replicative stress has received a lot of attention recently [23]. p53 regulates the expression of a wide number of target genes involved in cell cycle arrest, DNA repair, senescence and apoptosis when it is stimulated [24]. p53’s role in DNA damage response has been shown in numerous studies to be crucial in the maintenance of genomic integrity [25]. The loss of p53 function enhances chromosomal instability (directly and indirectly), causing cells to enter senescence or apoptosis [26]. It has been proved that the FoxO signaling pathway is a key factor in cell senescence mediated by Gas6 (growth stagnation specific protein) and plays a crucial multi-dimensional role in vascular aging and sclerosis [27]. These results indicate that ω-3 fatty acids can affect diseases and metabolic pathways related to human aging by affecting the expression of related genes. There is also some evidence that the proportion of maternal dietary fat as ω-3 PUFA, particularly α-linolenic acid, may also be associated with lower epigenetic age acceleration in newborns. In adults, accelerated epigenetic age is associated with an increased risk of cancer, cardiovascular disease and all-cause mortality [28]. In a new study recently published by researchers at Marshall University, researchers noticed significant differences in the offspring of mice whose mothers were fed a diet rich in canola oil, which is rich in ω-3 fatty acids, compared with mice whose mothers were fed a diet rich in corn oil, which is rich in ω-6 fatty acids. Maternal ω 3-rich diets influence genome-wide epigenetic pattern changes in offspring and may modulate gene expression patterns [29]. Therefore, we speculate that the effect of ω-3 fatty acids on aging may also be realized by influencing gene expression patterns. This genetic influence can be transmitted from early childhood to old age.

For the results of Mendelian randomization, currently, the study on the effect of *Bifidobacteria* on aging has been relatively sufficient. There is evidence that *Bifidobacteria* can regulate the treatment of specific age-related diseases and can be used as a treatment option for anti-aging. Oral administration of substrains of *Bifidobacterium* isolated from healthy centenarians can improve cell, body fluid and non-specific immune function and immune barrier function in mouse intestines, reduce inflammation, improve adaptive immune response and fight immune senescence [30]. Other regulatory mechanisms include regulating carbohydrate degradation, improving antioxidant activity, producing vitamin B and conjugated linoleic acid [31], regulating fat generation deposition and metabolism [32] and preventing insulin resistance [33]. In addition, these mechanisms also include improving the intestinal barrier function, reducing the production of short-chain fatty acids, increasing enzymes that have a significant impact on lipid metabolism and glucose homeostasis and limiting caloric intake [34]. Calorie restriction (CR) is currently one of the most feasible and effective anti-aging methods. CR can enrich genes that are positively related to longevity and reduce genes that are negatively related to longevity [35]. In a mouse model, oral administration of *Bifidobacterium* and *Lactobacillus Plantarum* can effectively prevent skin photoaging caused by chronic ultraviolet radiation [36,37]. In experimental studies, *Bifidobacteria* have also been shown to extend lifespan. It has been found in nematodes that supplementation with probiotics represented by *Bifidobacteria* has a significant effect on prolonging life. They promote longevity by stimulating an innate immune response [38], improving oxidative stress [39] and reducing lipofuscin accumulation [39]. There is also evidence that these probiotics can extend the lifespan of mice, possibly by inhibiting the chronic inflammatory process in the colon [40]. In addition, *Bifidobacteria* can also affect the senescence of the host by regulating the expression of some genes of the host [41].

For *Actinobacteria*, studies have shown that aging increases the number of actinobacteria [42]. For *Streptococcus*, Japanese researchers have proved that adding freeze-dried *Streptococcus* to the diet can inhibit aging [43].

On the other hand, reasonable dietary supplements can effectively intervene in the gut microbiome variation of humans and contribute to evaluating the causality between the gut microbiome and diseases [44]. In this study, the Mendelian randomization analysis for exploring the causal relationship between the influence of ω-3 fatty acids on blood lipids and the effect of gut microbiome variation proved that ω-3 fatty acids could have an impact on the gut microbiome, and the gut microbiome changing in group composition will have a direct impact on aging [45], indicating that part of the anti-aging mechanisms of ω-3 fatty acids is achieved through the gut microbiome.

Based on the multi-omics data related to ω-3 fatty acids, we performed a series of bioinformatics analyses, including methylation analysis, transcriptome analysis and Mendelian randomization. Our results imply that it has the effect of anti-aging and preventing aging-related disease. There are previous studies showing consistent results, such as a significantly reduced risk of ischemic events, including cardiovascular death, in patients with elevated triglyceride levels on statins compared with patients receiving placebo at 2 grams eicosapentaenoic twice daily [46]. However, several studies have found precisely the opposite, such as no significant difference in the compound outcome of major adverse cardiovascular events associated with the addition of ω-3 CA to conventional background therapy compared with corn oil in statin patients with high cardiovascular risk [47]. In another study exploring the effects of Marine n-3 fatty acids on cardiovascular disease and cancer, the supplementation of n-3 fatty acids did not reduce the incidence of major cardiovascular events or cancer compared with placebo [48]. In another study, dietary LA did play an important role in reducing cardiovascular risk. There is no evidence for its potential role in diabetes prevention [49].

These conclusions seem contradictory, and we speculate that the reason for this discrepancy is that there are differences in the types of people studied. Such differences may result in confounding assumptions and, thus, bias the results [50]. From the perspective of diseases, although some different types of diseases are associated with aging, the effects of the same nutrient on different diseases or different nutrients on the same disease are completely different [49]. Therefore, we recommend that in future studies of this kind, select as many different populations as possible as samples and, where possible, replicate the associations in databases with potentially confounding structures that differ from the initial study. Finding the same correlation in different populations may prevent confusion and misdirection [50].

In summary, our study provides deeper insights into the anti-aging mechanisms of ω-3 fatty acids based on the multi-omics data. The possible anti-aging mechanism of ω-3 fatty acids was analyzed using methylation analysis, transcriptome analysis and Mendelian randomization; these were supplemented with previous studies that focused on clinical observation, providing an important reference for subsequent medical research and human diet improvement.

## Figures and Tables

**Figure 1 genes-12-01691-f001:**
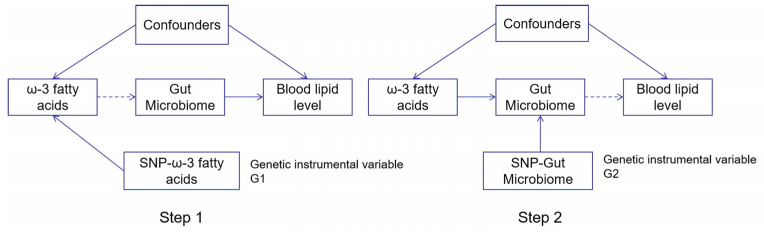
The flowchart showing the two-step Mendelian randomization methods used in this study. Step 1: Analysis of the association of ω-3 fatty acids on blood lipid levels using Single-nucleotide polymorphism (SNP) loci associated with ω-3 fatty acids as a proxy tool. Step 2: Analysis of the effect of the gut microbiome variation on blood lipid level using SNP loci associated with gut microbiome variation as a proxy tool for intermediate variables. Both genetic instrumental variables were independent of confounders. Appendix A show the detailed process results of MR analysis.

**Figure 2 genes-12-01691-f002:**
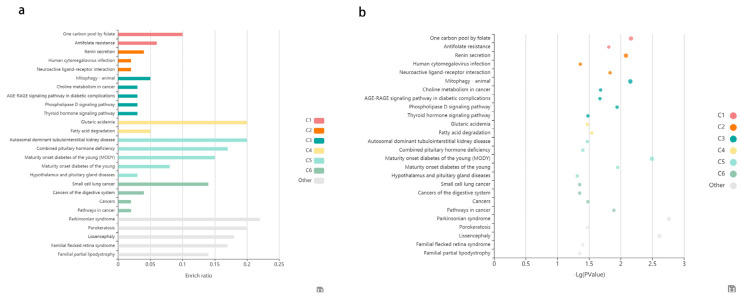
The enriched KEGG pathways of differentially methylated genes. (**a**) The enriched KEGG pathways of differentially methylated genes visualized in the bar plot. Each row represents an enriched function, and the length of the bar represents the enrich ratio, which is calculated as “input gene number”/” background gene number”. The color of the bar (C1–C6) represents different clusters, the terms in the same cluster have the similar functions. For each cluster, if there are more than 5 terms, the top 5 with the highest enrich ratio will be displayed. The enrichment analysis was conducted using KOBAS. (**b**) The enriched KEGG pathways of differentially methylated genes visualized in bubble plot. Each bubble represents an enriched function, and the size of the bubble from small to large: (0.05,1), (0.01,0.05), (0.001,0.01), (0.0001,0.001), (1 × 10^−10^, 0.0001), (0, 1 × 10^−10^). The color of the bubble (C1–C6) is the same as the color of the bar, which represents different clusters. For each cluster, if there are more than 5 terms, the top 5 with the highest enrich ratio will be displayed. The enrichment analysis was conducted using KOBAS.

**Figure 3 genes-12-01691-f003:**
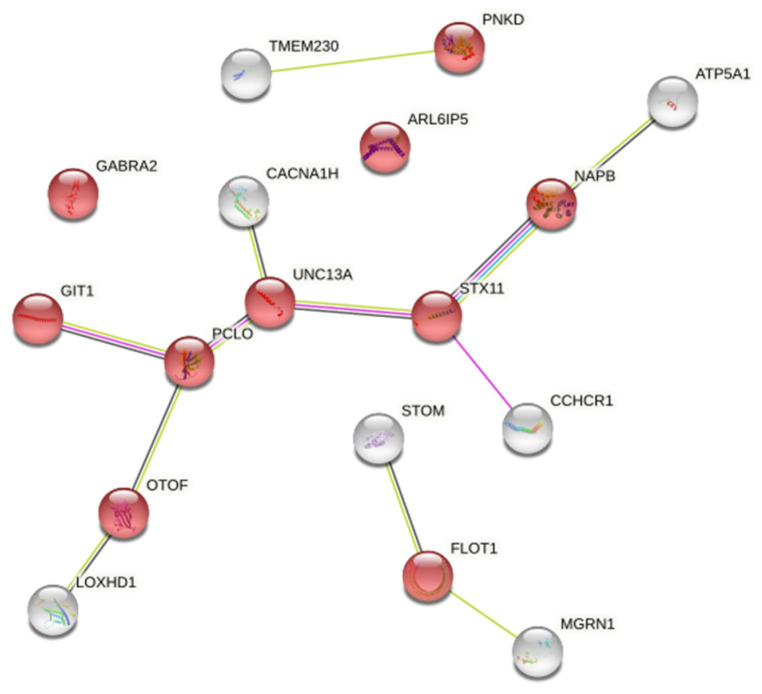
The ten proteins represented by red nodes were enriched in the biological process category of “neurotransmitter level enrichment regulation”. (The input genes were 140 methylated differential genes based on the differential methylation probe method). The abbreviations in the figure is input protein name and network nodes represent proteins and the splice isoforms or post-translational modifications are collapsed, i.e., each node represents all the proteins produced by a single, protein-coding gene locus. The white nodes represent the input proteins that interact with the red nodes.

**Figure 4 genes-12-01691-f004:**
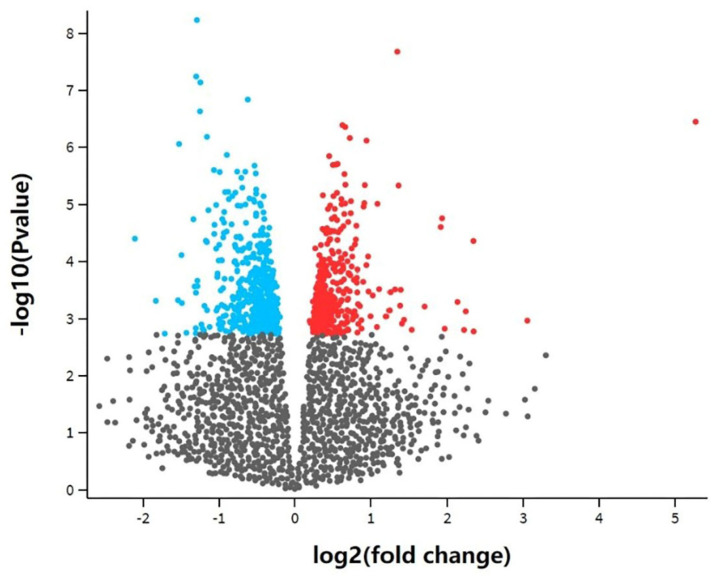
Differential gene volcano plot of transcriptome analysis. The blue part represents down-regulated genes, the red part represents up-regulated genes and the grey part represents non-differential genes. The differentially expressed genes were obtained using GEO2R. There was a total of 891 differentially expressed genes.

**Figure 5 genes-12-01691-f005:**
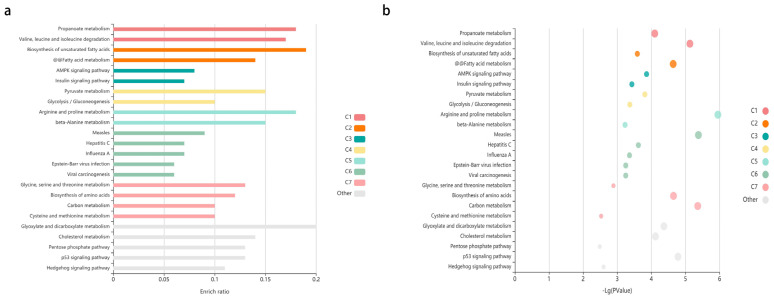
The enriched KEGG pathways of differentially expressed genes. (**a**) The enriched KEGG pathways of differentially expressed genes visualized in the bar plot. Each row represents an enriched function, and the length of the bar represents the enrich ratio, which is calculated as “input gene number”/”background gene number”. The color of the bar represents different clusters. For each cluster, if there are more than 5 terms, the top 5 with the highest enrich ratio will be displayed. The enrichment analysis was conducted using KOBAS. (**b**) The enriched KEGG pathways of differentially expressed genes visualized in bubble plot. Each bubble represents an enriched function, and the size of the bubble is from small to large: [0.05,1], [0.01,0.05], [0.001,0.01], [0.0001,0.001], [1 × 10^−10^,0.0001] and [0,1 × 10^−10^]. The color of the bubble is the same as the color of the bar above, which represents different clusters. For each cluster, if there are more than 5 terms, the top 5 with the highest enrich ratio will be displayed. The enrichment analysis was conducted using KOBAS.

**Figure 6 genes-12-01691-f006:**
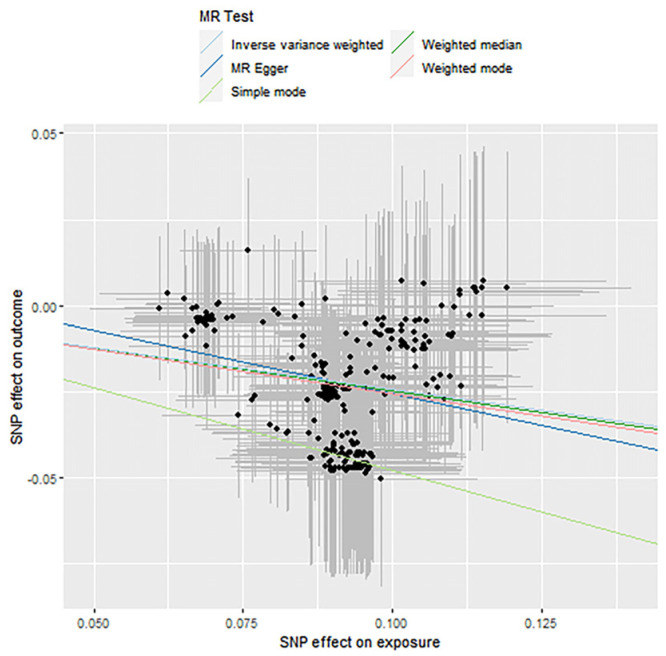
Causality analysis of *Actinobacteria* and blood lipid levels based on Mendelian randomization. The *x*-axis represents the effect value associated with SNP and *Actinobacteria*, and the *y*-axis represents the effect value associated with SNP and LDL and levels in the blood.

**Figure 7 genes-12-01691-f007:**
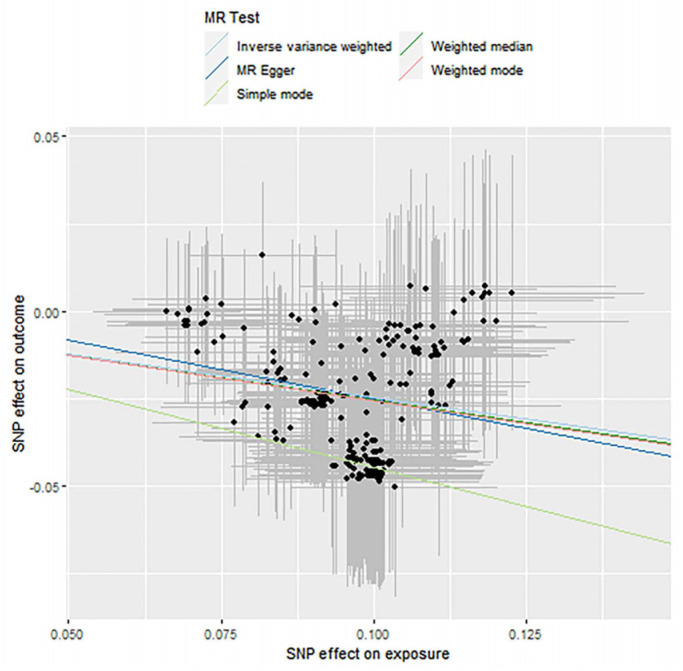
Causality analysis of *Bifidobacteria* and blood lipid levels based on Mendelian randomization. The *x*-axis represents the effect value associated with SNP and *Bifidobacteria*, and the *y*-axis represents the effect value associated with SNP and LDL and levels in the blood.

**Figure 8 genes-12-01691-f008:**
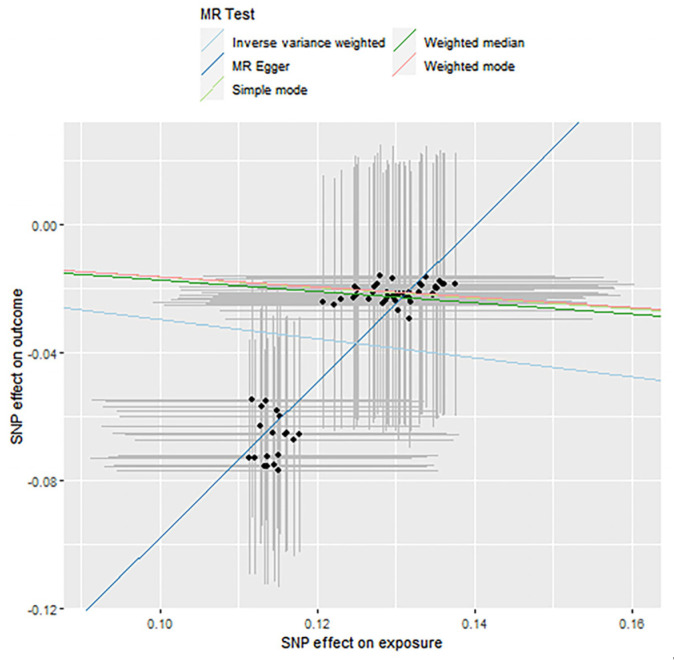
Causality analysis of *Streptococcus* and blood lipid levels based on Mendelian randomization. The *x*-axis represents the effect value associated with SNP and *Streptococcus*, and the *y*-axis represents the effect value associated with SNP and LDL levels in the blood.

**Table 1 genes-12-01691-t001:** The GO enrichment results of differentially methylated genes related to aging (based on SIMPO algorithm).

GO-Term	Description	Strength	False Discovery Rate
0000977	RNA polymerase II transcription regulatory region sequence-specific DNA binding	0.41	3.67 × 10^−2^
0000976	Transcription regulatory region sequence-specific DNA binding	0.39	3.67 × 10^−2^
1990837	Sequence-specific double-stranded DNA binding	0.39	3.67 × 10^−2^
0043565	Sequence-specific DNA binding	0.35	3.67 × 10^−2^

**Table 2 genes-12-01691-t002:** The KEGG enrichment results of differentially methylated genes related to aging (based on SIMPO algorithm).

Category	Term	*p*-Value	Related Genes
KEGG_PATHWAY	Signaling pathways regulating pluripotency of stem cells	1.53 × 10^−3^	*POU5F1* *WNT16* *AKT2* *RIF1* *ID1*
KEGG_PATHWAY	Cellular senescence	2.69 × 10^−^^3^	*AKT2* *RRAS2* *CCDC109A* *SIRT1* *RBBP4*
KEGG_DISEASE	Parkinsonian syndrome	1.74 × 10^−^^3^	*PINK1* *DNAJC13*
KEGG_DISEASE	Lissencephaly	2.44 × 10^−^^3^	*NDE1* *TMTC3*
KEGG_DISEASE	Maturity onset diabetes of the young (MODY)	3.26 × 10^−^^3^	*HNF1B* *PDX1*

**Table 3 genes-12-01691-t003:** The GO enrichment results of differentially methylated genes related to aging (based on differentially methylated probes).

	GO-Term	Description	Strength	False Discovery Rate
Biological Process	0007156	Homophilic cell adhesion via plasma membrane adhesion molecules	1.07	2.63 × 10^−6^
	0001505	regulation of neuro-transmitter levels	0.81	2.88 × 10^−2^
	0007399	Nervous system development	0.32	3.44 × 10^−2^
	0007275	multicellular organism development	0.23	3.35 × 10^−2^
Molecular Function	0005509	calcium ion binding	0.58	5.20 × 10^−4^

**Table 4 genes-12-01691-t004:** Enrichment analysis results of differentially expressed genes related to aging (based on transcriptome analysis).

Category	Term	*p*-Value
GAD_DISEASE	Alzheimer’s disease	3.38 × 10^−3^
GAD_DISEASE	Alzheimer’s Disease	4.59 × 10^−3^
GAD_DISEASE	longevity	6.88 × 10^−3^
GOTERM_CC_DIRECT	mitochondrion	1.61 × 10^−6^
GOTERM_CC_DIRECT	nucleolus	2.66 × 10^−4^
GOTERM_CC_DIRECT	mitochondrial matrix	7.42 × 10^−4^
KEGG_PATHWAY	p53 signaling pathway	4.75 × 10^−3^
KEGG_PATHWAY	FoxO signaling pathway	1.71 × 10^−2^

**Table 5 genes-12-01691-t005:** Mendelian randomization results of *Actinobacteria* and blood lipid levels.

Type	Category	Method	*p*-Value	OR
LDL	Class	MR Egger	7.47 × 10^−3^	6.94 × 10^−1^
LDL	Class	Inverse variance weighted	2.96 × 10^−42^	7.84 × 10^−1^
LDL	Class	Weighted median	1.90 × 10^−25^	7.80 × 10^−1^
TAGs	Class	MR Egger	9.87 × 10^−1^	9.98 × 10^−1^
TAGs	Class	Inverse variance weighted	2.51 × 10^−22^	1.19
TAGs	Class	Weighted median	1.11 × 10^−12^	1.18
LDL	Phylum	MR Egger	4.37 × 10^−1^	1.35
LDL	Phylum	Inverse variance weighted	8.11 × 10^−14^	7.70 × 10^−1^
LDL	Phylum	Weighted median	2.94 × 10^−10^	7.56 × 10^−1^

**Table 6 genes-12-01691-t006:** Mendelian randomization results of *Bifidobacteria* and blood lipid levels.

Type	Category	Method	*p*-Value	OR
LDL	Family	MR Egger	2.32 × 10^−2^	7.14 × 10^−1^
LDL	Family	Inverse variance weighted	4.08 × 10^−44^	7.81 × 10^−1^
LDL	Family	Weighted median	7.77 × 10^−26^	7.75 × 10^−1^
LDL	Genus	MR Egger	1.48 × 10^−1^	8.11 × 10^−1^
LDL	Genus	Inverse variance weighted	9.57 × 10^−43^	7.84 × 10^−1^
LDL	Genus	Weighted median	4.07 × 10^−23^	7.80 × 10^−1^
TAGs	Genus	MR Egger	7.34 × 10^−1^	9.53 × 10^−1^
TAGs	Genus	Inverse variance weighted	3.75 × 10^−22^	1.18
TAGs	Genus	Weighted median	1.12 × 10^−12^	1.18

**Table 7 genes-12-01691-t007:** Mendelian randomization results of *Streptococcus* and blood lipid levels.

Type	Category	Method	*p*-Value	OR
LDL	Genus	MR Egger	1.26 × 10^−4^	1.14 × 10^−1^
LDL	Genus	Inverse variance weighted	6.78 × 10^−14^	7.42 × 10^−1^
LDL	Genus	Weighted median	1.03 × 10^−3^	8.39 × 10^−1^

## Data Availability

The data presented in this study are available in Appendix A.

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
