# Peer review of "Multi-Omics Interpretation of Anti-Aging Mechanisms for ω-3 Fatty Acids"

_genes, 2021, doi:10.3390/genes12111691_

Round 1

Reviewer 1 Report

In this paper, where bioinformatics is the main component, terms such as GO analysis and network are used incorrectly and confuse the reader. It is important to point out to the authors that overrepresentation of biological annotations is the key task of any GO ENRICHMENT o r pathway ENRICHMENT analysis. The functional analysis flow for both methylation and expression is very confuse.

It is surprising to read the results for methylation and expression analysis separately, without trying to integrate results form both sources in the discussion.

The outcome of the work only provides a list of very general GO terms that are discussed and arbitrarily assigned as ageing related. Therefore this work doesnt bring any information about deeper insights into the anti-aging mechanisms of ω-3 fatty acids, it just describes some molecular functions associated with its supplementation on 2 experiments selected on GEO database.

1-Introduction

Authors should mention effect of fasting/caloric restriction on antiageing mechanisms

2- Methods

WHy to use SIMPO algorithm , is there any bibliographical reference to this method? 

What does NG16 and MBG stand for?

Results:

l176: which network are you referring to?

l190: what do you mean by "played a a suggestive role in the experimental results"?

l191: GO analysis is confusing, did you mean a GO enrichment analysis? Not clear

l194: you mean that you select genes annotated with GO terms linked to aging? Not clear

l202: the protein interaction network signal process we got

"We got" is not scientific language. Not clear which "protein interaction network signal process" you refer to

l204 no verb in the sentence

l272: volcano plot

greys are also down and up

l279: Then, the differentially expressed genes were enriched and analyzed in DAVID.

Are you enriching genes? This term is not appropiate here

Table 4: how come Nucleoplams is related to agein? Selection of age related terms is done without justification.

l 282: enriched in Alzheymer disease? Not apropiate here

Reviewer 2 Report

Multi-omics interpretation of anti-aging mechanisms for ω-3 fatty acids

In this study, the DHA-rich fish oil (800 mg DHA/d) or a placebo supplement was given to pregnant women from 20 weeks of pregnancy until delivery, and DNA methylation was analyzed from the samples of dried neonatal bloodstains via SIMPO algorithm, PANTHER, STRING, and KOBAS. Human placenta HTR8/SVneo cells were cultured with DNA-rich fish oil or soybean oil. Untreated cells were the control group. Transcriptome analysis was performed via GEO2R, DAVID, and KOBAS. The effect of fish oil on blood lipid level and the impact of the gut microbiome were determined by Mendelian randomization analysis. The results showed that ω-3 fatty acids can affect methylation levels and expression levels of genes associated with age-related diseases or pathways in humans and that ω-3 fatty acids on aging are partially mediated by the gut microbiome (including Actinobacteria, Bifidobacteria, and Streptococcus) causing changes in the levels of LDL and TAGs.

The problems are as follows:

  1. Major comments:
  2. In the study of pregnant women, other foods eaten by pregnant women may influence the methylation of fetus cells even though pregnant women ate more ω-3 fatty acids than the control group. A mice study is suggested to confirm the results of this study.
  3. The aging-related genes may have other functions at development. How do you link the expressed genes at neonate to the expressed genes at old age? For example, Line 390-393.
  4. In the study of human placenta HTR8/SVneo cells, the different expression of transcriptome depends on the DNA-rich fish oil or soybean oil per se. How do you link the expressed genes induced by these oils to the expressed genes at old age?
  5. Line 202-213, calcium is a second messenger; therefore, all cellular functions are regulated by calcium. Why do you limit the calcium effect on the nervous system? How do you define calcium concentration as related to normal cellular functions or diseases?
  6. The expressed genes directly or indirectly induced by ω-3 fatty acids should be explained and previous studies supported the gene assays of this manuscript should be discussed such as in calcium level, cadherins system, and nervous system development.
  7. The different expression of the gut microbiome depends on the DNA-rich fish oil or soybean oil per se. How do you link the three types of the gut microbiome's strong association with the aging of hosts?
  8. Numerous previous studies of ω-3 fatty acids should be discussed such as follows to increase the content of the Discussion.
  9. 1. Nicholls et al. JAMA.2020;324:2268-2280.
  10. 2. Manson et al. Engl. J. Med. 2019;380:23-32.
  11. Bhatt et al. N. Engl. J. Med. 2019;380:11-22.
  12. Schulze et al. Lancet Diabetes Endocrinol. 2020;8:915-930.
  13. Bischoff-Ferrari et al. JAMA. 2020;324:1855-1868.
  14. Tu et al. Nutrients. 2020;12:3301.

Minor comments:

  1. Line 39, please provide more references.
  2. Ref3 and Ref 5 are similar. Ref3 is right. Ref5 is wrong. Remove Ref5.
  3. Line 55-56, please cite references.
  4. Line 71, what is the reason that 800 mg DHA/d is used?
  5. Line 394-396, these phenomena may be related to energy utilization of ω-3 fatty acids.

Reviewer 3 Report

Using multiomics data the authors tried to find out more about the anti-aging effects of n-3 polyunsaturated fatty acids. While I think the contribution might be a useful read for people in the field, it required substantial improvement. I) please provide a new structure and add subchapters outlining the content. Ii) follow a logical order in your arguments so that the reader receives the context easily. Iii) find my editorial suggestions below and iv) have a native speaker read the paper before re-submission to remove all language-based errors.

Page 1, Line 10: change to „previous research suggested….”

Page 1, line 17: change to “ on blood lipid levels and variation in the gut microbiome”

Page 1, line 26: which refers to gradual degenerative changes (no “the”)

Page 1, line 27: replace mortality by “frailty”

Page 1, line 28: end sentence with increasing age (delete until …)

Page 1, line 40: replace by “has an effect on preventing aging”

Page 2, line 58: replace thinking loss by “cognitive losses”

Page 2, line 59: delete “also”

Page 2, line 73: Rephrase. Genome methylation data at birth from 369 children were obtained….

Page 3, line 96: change to “ The “minfi” package from R (https://www.r-project.org/) was used to…”

Page 3, line 106: Change the sentence to “The genes were then entered on to the STRING website for GO analysis”.

Page 3, line 129: change to “ to have a p-value of less than…”

Page 4, line 141: MR hypothesis? Please explain and elaborate on it

Page 4, line 158: change to “The advantage of comparing the results from three different approaches is higher reliability due to an increased consistency”

Page 5, line 180: change the tense from “is” to “there was”

Page 5, line 188: change to “all genes were from Homo sapiens”

Page 5, Line 191: “played an informative role in the results”

Page 5, Line 194: change to “ Then we entered all those….genes”

Page 5, line 203: change is to “was” and use past tense-> was mainly associated

Page 5, line 203: calcium combination sounds awkward. Do you mean calcium metabolism/calcium turnover?

Page 5, line 204: in the nervous system (there is no neuron system)

Page 5, line 204: rephrase the entire sentence since there is a verb missing and there are several mistakes.

Page 5, line 210. Above you write about Parkinsonian syndrome now you mention it as disease. For consistency please choose one and keep it

Page 5, line 210: Please rephrase the sentence it is very difficult to understand

Page 5 line 217: pls. rephrase. A verb is missing and the meaning is unclear

Page 6, line 221: change to “catenin proteins“

Page 6, line 223: replace “homophile manner” since this is misleading and rephrase

Page 6, line 227: cadherin system

Page 6, line 228: replace neuron cell

Page 7, line 260: “represents the enrichment ratio” (enrich doesn’t mean anything)

Page 8, Line 272: a volcano map was drawn/constructed/generated

Page 9, Line 303: on blood lipid levels and the influence of variation in the gut microbiome

Page 9, line 311: corresponding (no capital letter)

Page 9, line 318: remove the reference here and discuss it in the discussion chapter.

Page 10, line 321: what do you mean by “relatively sufficient” (good/bad)?

Page 10, line 323-348: these are no suitable sentences for a results chapter since the refer to other people’s work. Please save this info for the discussion

Page 12, line 369: analyses (plural)

Page 12, line 380: human aging is related to neural development??? I’d say rather the opposite, degenerative processes of neurons

Round 2

Reviewer 1 Report

l.373: and the grey part represents non-differential genes

I would rather state that the grey genes are not statistically significant

Author Response

Thank you for your suggestion. We have modified the expression according to your instruction (line 387).

Reviewer 2 Report

If authors cannot provide new results, I recommend rejecting this manuscript because these results may deliver misleading information currently. The problems are as follows:

Major comments:

  1. Although the authors explained that the purpose of this study is to provide a preliminary interpretation of the anti-aging mechanism for ω-3 fatty acids using bioinformatics methods; however, I am concerned that misleading information may be delivered due to imprecise methods. Especially, the neonatal bloodstains were used to answer the anti-aging mechanism. The cells of neonatal bloodstains were mixed with the young cells, necrosis cells, and old cells of many different cell types. Thus, the results from neonatal bloodstains were derived from many variables. Besides, the use of human placenta HTR8/SV neo cells did not show the population doublings curves to collect young and aged cells and did not mention the duration and concentration of oil treatment. Thus, the results on human placenta HTR8/SV neo cells are ambiguous. Although authors obtained the data of gene expression under these methods, I suspect the value of these data. Therefore, I strongly recommend that authors should provide the results of the mice study in this manuscript because the variable in the mice study is simple and can provide credible evidence. If the results of mice support those of human fetus cells and human placenta HTR8/SV neo cells, these results would be convincing.
  2. The questions of Point2 and 3 still exist.
  3. In Abbas’s study, they mentioned that omega 3 fatty acids have a function to prevent cancer development (Abbas et al., 2021). However, the gene expression of preventing cancer development and aging may be similar, but anti-aging may be different.

Author Response

Thank you again for your review. We will answer your questions from the following three aspects.

  • First, you think it is not appropriate to use the neonatal bloodstains to answer the anti-aging mechanism. And you supposed results from neonatal bloodstains may be derived from many variables.

Actually, we also have concerned about these problems. We have read the original article that provides these data, and found that their purpose is to evaluate the effect of prenatal DHA supplementation on the infant epigenome. Therefore, we supposed these data are feasible and convincing. There was also some evidence that the proportion of maternal dietary fat as n–3 PUFAs, in particular, α-linolenic acid, may be associated with lower epigenetic age acceleration in the newborn, and such age acceleration begins before birth [1]. However, we also agree with you that the use of the term "anti-aging" is not so accurate and may cause some misunderstandings among readers. Therefore, we revised some "anti-aging" in the new manuscript to "delay aging". On the other hand, this neonatal bloodstains data are the only appropriate data we could acquire from the public database. After careful consideration, we think these data are appropriate.

[1] Phang M, Ross J, Raythatha J H, et al. Epigenetic aging in newborns: role of maternal diet[J]. The American journal of clinical nutrition, 2020, 111(3): 555-561.

  • Second, you mentioned the use of human placenta HTR8/SV neo cells did not show the population doublings curves to collect young and aged cells and did not mention the duration and concentration of oil treatment. Thus, the results on human placenta HTR8/SV neo cells are ambiguous.

For this question, we are really sorry, because it is our oversight that hasn’t put the detailed data information into our manuscript, we have added the relevant information in our method part to make the results more accurate (line79~line89).

  • Third, you recommend that we should provide the results of the mice study in this manuscript. We are really sorry that we are not able to conduct such a mice study at the present stage. And we would make it our further plan.

[Point 2] The questions of Point 2 and 3 still exist.

[Response]

[Point 2] The aging-related genes may have other functions at development. How do you link the expressed genes at neonate to the expressed genes at old age?

[Point 3] In the study of human placenta HTR8/SVneo cells, the different expression of transcriptome depends on the DNA-rich fish oil or soybean oil per se. How do you link the expressed genes induced by these oils to the expressed genes at old age?

Sorry, we haven’t explained clearly to you in the last round of modification. In our study, the direct association between the ω-fatty acids and expressed genes at old age may be hard to be acquired. So the concept of aging here we are focusing on is actually “Epigenetic aging”, which could even be existing in newborns [1]. And these Epigenetic “clocks” may now surpass chronological age in accuracy for estimating biological age [2].

[1] Phang M, Ross J, Raythatha J H, et al. Epigenetic aging in newborns: role of maternal diet[J]. The American journal of clinical nutrition, 2020, 111(3): 555-561.

[2] Fahy, Gregory M., et al. "Reversal of epigenetic aging and immunosenescent trends in humans." Aging cell 18.6 (2019): e13028.

[Point 3] In Abbas’s study, they mentioned that omega 3 fatty acids have a function to prevent cancer development (Abbas et al., 2021). However, the gene expression of preventing cancer development and aging may be similar, but anti-aging may be different.

[Response] Thanks for your review. Yes, you are right, the word “anti-aging” may not be so accurate and could cause some misunderstanding. We have modified relevant expressions in our new manuscript to make it more accurate. Actually, our purpose in this part is not to provide evidence that the differentially expressed genes may target anti-aging genes. We were just trying to find some evidence that ω-3 fatty acids may influence the expression of aging-related genes.

Reviewer 3 Report

The authors have dealt with my revisions with due diligence. Thank you for addressing each point. The referencing of the R packages could be improved and some english editing is required ahead publication.

Author Response

Thanks for your advice. We have improved the referencing of R packages (line126; line179) and have our manuscript checked by a native English-speaking colleague.